# The Impact of Sciatic Nerve Injury on Extracellular Matrix of Lower Limb Muscle and Thoracolumbar Fascia: An Observational Study

**DOI:** 10.3390/ijms25168945

**Published:** 2024-08-16

**Authors:** Xiaoxiao Zhao, Caterina Fede, Lucia Petrelli, Carmelo Pirri, Elena Stocco, Chenglei Fan, Andrea Porzionato, Cesare Tiengo, Raffaele De Caro, Stefano Masiero, Carla Stecco

**Affiliations:** 1Institute of Human Anatomy, Department of Neurosciences, University of Padova, 35121 Padova, Italy; 2Padova Neuroscience Center, University of Padova, 35129 Padova, Italy; 3Department of Women’s and Children’s Health, University of Padova, 35128 Padova, Italy; 4Department of Surgery, Oncology and Gastroenterology, University of Padova, 35124 Padova, Italy; 5Plastic and Reconstructive Surgery Unit, Department of Neuroscience, University of Padova, 35128 Padova, Italy; 6Neurorehabilitation Unit, Department of Neuroscience, General Hospital—University of Padova, 35121 Padova, Italy

**Keywords:** extracellular matrix, peripheral nerve injury, hyaluronan, collagen, intramuscular connective tissue, thoracolumbar fascia

## Abstract

Peripheral nerve injury (PNI) is a complex clinical challenge resulting in functional disability. Neurological recovery does not always ensure functional recovery, as extracellular matrix (ECM) alterations affect muscle function. This study evaluates hyaluronan (HA) and collagen concentration in the gastrocnemius muscle and thoracolumbar fascia (TLF) in unilateral lower limb PNI rats to explore systemic ECM alterations following PNI and their impacts on functional recovery. Eighteen 8-week-old male Sprague-Dawley rats were divided into experimental (*n* = 12 left sciatic nerve injury) and control (n = 6) groups. After six weeks, motor function was evaluated. Muscle and TLF samples were analysed for HA and collagen distribution and concentrations. SFI and gait analysis confirmed a functional deficit in PNI rats 6 weeks after surgery. HA concentration in both sides of the muscles decreased by approximately one-third; both sides showed significantly higher collagen concentration than healthy rats (12.74 ± 4.83 µg/g), with the left (32.92 ± 11.34 µg/g) significantly higher than the right (20.15 ± 7.03 µg/g). PNI rats also showed significantly lower HA (left: 66.95 ± 20.08 µg/g; right: 112.66 ± 30.53 µg/g) and higher collagen (left: 115.89 ± 28.18 µg/g; right: 90.43 ± 20.83 µg/g) concentrations in both TLF samples compared to healthy rats (HA: 167.18 ± 31.13 µg/g; collagen: 47.51 ± 7.82 µg/g), with the left TLF more affected. Unilateral lower limb PNI induced HA reduction and collagen accumulation in both the lower limb muscles and the TLF, potentially exacerbating motor function impairment and increasing the risk of low back dysfunctions.

## 1. Introduction

Peripheral nerve injury (PNI) is a challenging clinical issue with diverse range of symptoms, including muscle weakness, pain, and paraesthesia, often resulting in functional disability and affecting quality of life [1,2,3]. The most severe degree of PNI is neurotmesis, a nerve transection injury with complete disruption of endoneurium, perineurium, and epineurium [3]. Nerve transection injuries with a substance loss exceeding 3 mm often require surgical intervention by using nerve grafts for neurological recovery. However, neurological recovery does not guarantee functional recovery resulting from nerve regeneration at the injury site or reinnervation and muscle function recovery of the effector’s end plate (muscle innervated by the injured nerve) [2]. After muscle denervation caused by neurotmesis, muscle fibres degenerate and are replaced by fibrous connective tissue, resulting in function loss [4]. Muscle function is determined not only by the muscle fibres that generate force but also by the intramuscular connective tissue (IMCT), which transmits force, regulating muscle flexibility and stiffness through its extracellular matrix (ECM) components [5,6].

Previous studies reported collagen accumulation in the denervated muscle ranging from 2 days to 5 weeks after denervation [7,8,9,10,11]. Indeed, the ECM of IMCT is composed of not only protein fibres such as collagen fibres, which provide a three-dimensional scaffold for muscle fibres, playing a crucial role for the force transition [5,6], but also ground substances regulating the gliding movement [12], composed of proteoglycans, multiadhesive glycoproteins, glycosaminoglycans, and water. Among these components, hyaluronan (HA) plays a vital role for the integrity of ECM [13]. HA is a high-molecular-weight glycosaminoglycan found in various tissues and fluids providing mechanical stability and serving as a water reservoir and lubricant [14]. The primary mechanical function of HA is to maintain viscosity to facilitate gliding between the two adjacent surfaces of fascia or IMCT and muscle, as well as between fascia sublayers [15,16,17]. The alterations in the concentration, molecular weight, or covalent modification of HA, and the changes in its binding interactions with other macromolecules, can significantly impact the gliding movement of IMCT or fascia [18], leading to various pathological conditions including pain and muscle dysfunction [19].

HA alteration has been widely observed in IMCT in skeletal muscle in different pathological conditions: an accumulation of HA was found in rats with immobilised joints [20] and in stroke patients [21], whereas a reduction of HA level and increasing collagen level was found in aged human and mouse muscles [22] as well as in patients with hip osteoarthritis [23]. However, HA in IMCT of muscles following nerve injury remains unexplored. Given that the gastrocnemius muscle is innervated by the tibial branch of the sciatic nerve, making it commonly used in animal studies of sciatic nerve injury [24,25,26], the primary goal of this study is to analyse the HA amount modifications in IMCT of the gastrocnemius muscle following sciatic nerve injury in rats.

The secondary goal of this study is to investigate ECM alterations in the gastrocnemius muscle on the contralateral side of left sciatic nerve injury rats. Previous animal studies demonstrated that unilateral lower limb PNI has a significant impact on lower limb muscle atrophy, affecting not only the injured side but also the contralateral side in a rat PNI model [27,28]. Specifically, two weeks post left L5 nerve injury, there was a significant decrease in muscle mass in the right gastrocnemius muscle, accompanied by reductions in myofibrillar protein content and cross-sectional area of type II fibres in the right plantaris muscle of the rats [27,28]. However, the changes of ECM on the muscle of the contralateral side after unilateral lower limb PNI remain unclear.

Thirdly, we also extend our investigation of ECM alterations in the thoracolumbar fascia (TLF), as evidence has shown a connection between the TLF and the fascia of the lower limb [29]. The aim is to explore whether the impact of a unilateral lower limb PNI is localised exclusively to the effector’s end plate—affecting the muscle innervated by the injured nerve or extending systemically—affecting the distally lower limb on the contralateral side and reaching proximally to the TLF.

Therefore, this study aims to investigate the concentration of HA and collagen in IMCT of the muscle innervated by the injured never, the muscle on the contralateral side, and in the TLF of the unilateral lower limb PNI rats to provide new insights into the localised and systemic effects of PNI on ECM.

## 2. Results

### 2.1. Motor Function

At 6 weeks from surgery, the sciatic functional index (SFI) score was calculated (the representative footprints image is showed in Appendix A), and the mean value was −87 ± 9.12 (a score of 0 indicates normal motor function of the sciatic nerve, while −100 indicates complete motor dysfunction of the sciatic nerve), indicating a severe deficit of motor function following the unilateral lower limb PNI. As for the gait analysis, the mean score was 1.32 ± 0.29 (maximum score is 2.0), revealing abnormal movement patterns. SFI and gait analysis confirmed functional deficit of rats 6 weeks after surgery.

### 2.2. Distribution of HA 

HABP (hyaluronic acid binding protein) immunohistochemistry staining showed that HA was localised in IMCT (epimysium, endomysium, and perimysium) in the gastrocnemius muscle (Figure 1A–C) and the TLF (Figure 2A–C) of the rats. Alcian Blue staining showed that abundant glycosaminoglycans uniformly distributed were found in IMCT in the gastrocnemius muscle (Figure 1D–F) and the TLF (Figure 2D–F) of the rats.

### 2.3. Distribution of Collagen

Picrosirius Red staining of the gastrocnemius muscle showed that collagen (in red, Figure 3) was localised in IMCT (epimysium, endomysium, and perimysium). In the injured leg (Figure 3A,D), the accumulation of collagen was greater than the controls (Figure 3C,F). Picrosirius Red staining of the TLF (Figure 4) of the rats showed that the TLF is rich in collagen. 

### 2.4. Quantification of HA in the Muscle and TLF

The HA concentration in the gastrocnemius muscle was as follows: 81.19 ± 33.53 µg/g on the left (sciatic nerve injury) side and 86.77 ± 35.25 µg/g on the right (contralateral) side of the left sciatic nerve injury rats and 127.57 ± 40.28 µg/g in the healthy rats (Figure 5A). The results showed no significant difference in the HA concentration in the gastrocnemius muscle between the left and right sides (*p* = 0.893) of the injured rats, whereas when compared to the healthy rats, the HA concentration of both the left and right sides of the injured rats were significantly lower (left: *p* = 0.013, right: *p* = 0.030) (Table 1).

The HA concentration in the TLF was as follows: 66.95 ± 20.08 µg/g on the left (sciatic nerve injury) side and 112.66 ± 30.53 µg/g on the right (contralateral) side of the left sciatic nerve injury rats and 167.18 ± 31.13 µg/g in the healthy rats (Figure 5B). The results indicated that the HA concentration in the TLF on the left side was significantly lower than that on the right side of the left sciatic nerve injury rats (t(11) = −7.31, *p* < 0.001). Similarly, when compared to the control group (healthy rats), the HA concentration in both the left side (t(16) = −8.32, *p* < 0.001) and right side (t(16) = −3.55, *p* = 0.003) of the TLF in the injured rats was also significantly lower (Table 1).

### 2.5. Quantification of Collagen in the Muscle and TLF

The collagen concentration in the gastrocnemius muscle was as follows: 32.92 ± 11.34 µg/g on the left (sciatic nerve injury) side and 20.15 ± 7.03 µg/g on the right (contralateral) side of the left sciatic nerve injury rats and 12.74 ± 4.83 µg/g in the healthy rats in the control group (Figure 6A). The results showed that the collagen concentration in the gastrocnemius muscle on the left side was significantly higher than that on the right side of the left sciatic nerve injury rats (*p* = 0.01). Compared to the control group (healthy rats), the collagen concentration of both the left (*p* < 0.001) and right (*p =* 0.022) sides of the injured rats is significantly higher (Table 1).

The collagen concentration in the TLF was as follows: 115.89 ± 28.18 µg/g on the left (sciatic nerve injury) side and 90.43 ± 20.83 µg/g on the right (contralateral) side of the left sciatic nerve injury rats and 47.51 ± 7.82 µg/g in the healthy rats (Figure 6B). The results indicated that the collagen concentration in the TLF on the left side was significantly higher than that on the right side of the left sciatica nerve injury rats (*p* = 0.027). Compared to the control group (healthy rats), the collagen concentration of the TLF on both the left (*p* < 0.001) and (right t(*p* < 0.002) sides of injured rats was significantly higher (Table 1). 

### 2.6. Area Percentage of Collagen Content in IMCT in the Gastrocnemius Muscle

The area percentage (area %) of collagen calculated based on Picrosirius Red staining images showed that the area % of collagen content in the gastrocnemius muscle is 28.01 ± 5.78% on the left (sciatic nerve injury) side and 17.33 ± 6.52% on the right (contralateral) side of the injured rats and 5.30 ± 0.82% in the healthy rats in the control group (Figure 7). The results showed that the collagen area % in the gastrocnemius muscle on the left side is significantly higher than that on the right side of the left sciatic nerve injury rats (*p* = 0.021). Compared to the control group (healthy rats), the collagen concentration of both sides of the injured rats was significantly higher (left: *p* = 0.021, right: *p =* 0.021) (Table 1). The original images for image analysis were shared in Figshare (DOI: 10.6084/m9.figshare.26535388). The area % of collagen content of the TLF is not calculated in this study due to the nature of the TLF, which is composed of collagen. 

## 3. Discussion

In this study, six weeks after the left sciatic nerve transection injury was repaired by nerve grafts, the SFI values of the rats in the experimental group were near −100, indicating a severe deficit of motor function following the unilateral lower limb PNI, as the scores range from 0 (normal function) to −100 (complete dysfunction) [30]. Gait analysis supports this functional deficit, revealing abnormal movement patterns. Our previous research revealed that six weeks postsurgery, OxPVA-based nerve grafts significantly increased the number and density of myelinated axons in the sciatic nerve transection injury rats [31] but were not sufficient enough to achieve functional recovery, which was probably hindered by the alterations in the ECM.

Our findings are the first to demonstrate that 6 weeks after an injury to the unilateral sciatic nerve, compared to the healthy rats, HA concentration in the IMCT of gastrocnemius muscle of both the left and right sides (sciatic nerve injury side and contralateral side) decreased by approximately one-third, from 127.57 µg/g to 81.19 µg/g and 86.77 µg/g; additionally, a greater reduction in HA was observed in the TLF on the injured side than on the contralateral side. In terms of collagen, both biochemical assays and analysis of collagen area % showed a significant increase in collagen levels (roughly one-third) within the IMCT of the gastrocnemius muscle on the sciatic nerve injury side compared to the contralateral side. This accumulation of collagen concentration aligns with the previous findings in the denervated muscles on rat models [7,8,9,10,11], but our study extends the observations to collagen accumulation on the contralateral side and at a distance in the TLF. Compared to the healthy rats, collagen levels increased nearly three times on the sciatic nerve injury side and two times on the contralateral side in both the gastrocnemius muscle and TLF. Combined with the HA alterations in both the muscle and TLF, these results indicate a systemic ECM alteration after unilateral lower limb peripheral nerve injury.

HA, the main component of the ground substance in the ECM, not only impacts the viscoelasticity of the ECM and facilitates the gliding among muscles [17,29] but also is critical for effective tissue repair and regeneration [32,33]. In muscle denervation following sciatic nerve transection injury, the reduced HA levels can limit the muscle repair and regeneration process [34,35], leading to compromised muscle functional recovery. On the contralateral side, the lower HA level could lead to a reduction in IMCT hydration, affecting lubrication and then altering the normal gliding of muscle during movement [22], which can influence muscle function. 

On the other hand, collagen is the most abundant component of the ECM, playing a crucial role in providing structural support for cells, and scaffolding for muscle fibres, and resistance to force, tension, and stretch to protect muscle fibres [36]. In general, the increased collagen concentration can increase the stiffness of the IMCT and muscle [37], which limits force transmission, leading to decreased functional capabilities and diminishing functional efficiency [38]. While the increased collagen level in the left gastrocnemius muscles of left sciatic nerve injury rats can be a protective factor for muscle atrophy during the stage of denervation [10], it limits the functional recovery after reinnervation and contributes to decreased elasticity and muscle mechanical performance.

Therefore, for functional recovery, a reduced HA level and increased collagen of the gastrocnemius muscle of both sides may aggravate these functional impairments by affecting the viscoelastic properties [18] and stiffness [37] of the gastrocnemius muscles, which are crucial for plantar flexion during walking [39,40].

The reduction in HA level and the concurrent increase in collagen concentration in unilateral lower limb PNI rats was also observed at a distance in the TLF. Compared to the healthy rats, the significant HA reduction observed on both the left and right sides in the left sciatic injury rats exceeded one-half on the left side and one-fourth on the right side, while the collagen level increased nearly three times on the injured side and two times on the contralateral side. Anatomically, the TLF serves as an attachment point for both the upper and lower limb muscles; functionally, it is essential for the efficient transfer of loads between the trunk and the limbs [41]. Structurally, the TLF is a thick, bandlike, multilayered structure consisting of three layers of collagen fibres separated by a thin layer of loose connective tissue [42,43], rich in HA, that facilitates the gliding of adjacent sublayers to maintain the normal function of the TLF [17]. The reduction in HA demonstrated in this study can reduce the gliding movement of the TLF, while an increase in collagen level can increase its stiffness. This combination can make the TLF become more rigid and stiffer. 

For overall function, these ECM alterations in the TLF can also contribute to abnormal gait. A human study suggested that the increased stiffness in the TLF, together with increased plantar-flexion resistance, could influence biomechanics of plantar flexion during walking [44] and therefore potentially impact gait patterns. In addition, clinical evidence has demonstrated that individuals with chronic low back pain (LBP) can experience reduced shear strain of the TLF [45]. The reduction in HA level demonstrated by this study can lead to a decrease in the gliding capabilities of the layers of connective tissue of the TLF [18], which can decrease the shear strain of the TLF with a potential increase in the risk of lower back pain [46]. At the same time, the increased collagen level in the TLF can increase the stiffness of the TLF. This stiffness, combined with reduced gliding due to HA reduction, may further exacerbate shear strain [46] and thereby potentially contribute to lower back pain and dysfunction. 

The ECM alterations demonstrated in this study underline the systemic implications of unilateral lower limb PNI, extending beyond the site of the effector’s end plate to the broader connective tissue network, potentially impacting overall musculoskeletal health and mobility. As fascia exhibits a remarkable capacity for remodelling in response to biomechanical stimuli [47], the observed ECM alterations in the IMCT in the gastrocnemius muscle of the contralateral side and TLF of the left sciatic nerve injury rat may not come directly from the nerve injury itself but rather from compensatory remodelling following the function loss of the injured limb after PNI. 

This study provides novel insights into systemic ECM alterations following unilateral lower limb PNI. Although this study is based on an animal model, the results can be translated to humans, suggesting that nerve injuries may have more widespread impacts than previously understood. A comprehensive assessment [48] and holistic rehabilitation approach in clinical practice are therefore essential so that clinicians can improve functional recovery and reduce the risk of secondary complications, leading to better overall outcomes for patients with PNI. 

### Limitations and Further Research

This study only evaluated the concentration of the HA but did not address its molecular weight, which is also important for HA’s function. Future research should also consider changes in other important ECM elements, such as elastic fibres. While total collagen content was assessed, future studies should separately investigate types of collagens, such as collagens I and III. Additionally, the sample size of the control group can be increased to the same level as the experimental group to increase the statistical power of the study. Moreover, as an observational study, this research cannot reveal the underlying mechanisms of ECM changes. Future mechanistic studies are needed to investigate the roles of interacting factors such as inflammatory, mechanical, and cellular responses in ECM remodelling through controlled experiments. This will help clarify their specific contributions and interactions. This study evaluated HA and collagen concentrations only at six weeks postinjury. Implementing longitudinal studies to monitor ECM changes at multiple time points will provide insights into the temporal dynamics of ECM remodelling and identify critical windows for therapeutic intervention.

## 4. Materials and Methods

### 4.1. Animal Model of PNI

Eighteen male Sprague-Dawley rats (8-week-old) were randomly allocated into two groups. Specifically, n = 12 animals underwent left sciatic nerve transection injury repaired by OxPVA-based conduit (experimental group); n = 6 animals represented the healthy control group. 

For the experimental group, following the induction of anaesthesia with a gas mixture of isoflurane and oxygen, the left thigh of each rat was shaved and disinfected. After that, a gluteal-splitting incision was performed to expose the sciatic nerve, which was then transected to create a 5 mm gap between the proximal and distal nerve ends. Thus, an OxPVA-based conduit (10 mm in length) was coaxially interposed between the nerve stumps and sutured to the epineurium with 8-0 nylon stitches to bridge the severed nerve (Figure 8). The surgical site was subsequently closed in a layered fashion using 4-0 silk sutures to ensure proper healing. Following surgery, the rats were allowed to recover in their cages and were housed within a temperature-regulated facility; comprehensive postoperative care was administered, including anti-inflammatory and antibiotic treatments (Rimadil, 5 mg/kg and Bytril, 5 mg/kg, respectively) for a duration of 5 days. The animals were provided with a standard laboratory rodent diet and had unrestricted access to water. The animals’ well-being was consistently monitored, encompassing assessments of their typical activities, uninterrupted feeding routines with no weight loss (the weight of each rat at weeks 1, 3, and 6 are reported in Appendix A), and the absence of any signs of wound infection or illness. 

Animal surgery and husbandry were performed following the Italian guidelines on the use of experimental animals and approved by the Ethical Committee of the University of Padua and by the Italian Department of Health (authorisation no. 837/2019-PR, 9 December 2019). 

### 4.2. Behavioural Tests of Motor Function

#### 4.2.1. Sciatic Functional Index

Six weeks postsurgery and prior to euthanasia, the motor function recovery was evaluated through a reliable and widely used noninvasive method based on footprint analysis and consisting of SFI calculation [49,50]. Briefly, in accordance with the procedure described previously [31,51,52], all the rats’ hindfeet were stained with black ink, after which the footprints were captured as they walked in a corridor lined with white paper. The SFI score was calculated considering specific measurements on both the operated-side paw footprint and the contralateral one to fulfil the formula established by Bain et al. [30]:SFI = [−38.3 × (EPL − NPL)/NPL] + [109.5 × (ETS − NTS)/NTS] + [13.3 × (EIT − NIT)/NIT] − 8.8

The following values were gained from the footprints: print length (PL), which measures the distance from the heel to the tip of the third toe; toe spread (TS), which measures the distance from the first to the fifth toe; and intermediary toe (IT) spread, which measures the distance between the second and fourth toes. Specifically, NPL, NTS, and NIT represent the PL, TS, and IT recorded from the nonoperated foot, respectively. EPL, ETS, and EIT represent the PL, TS, and IT recorded from the operated foot (E, experimental side), respectively. A score near 0 indicates normal sciatic nerve function, while a score approaching −100 indicates complete motor dysfunction of the sciatic nerve.

#### 4.2.2. Gait Analysis 

After footprint recording, the gait analysis was performed. In accordance with the procedure described previously [31], each rat was guided along a 50 cm long and 10 cm wide transparent plexiglass lane, with one end darkened. A mirror was positioned at a 45-degree angle underneath the lane to allow for the concurrent observation of the lateral profile of the rat throughout the gait cycle. The gait of each rat was recorded for analysis. The operated-side paw/limb was observed for toe spread, plantar walking, normal swing phase, smooth walking, absence of paw dragging, inversion and eversion, step alternation, hindfoot remaining in the body perimeter, and joint contraction [53]. Each parameter was scored through the following scale: 0 for nonassessable, 1 for assessable but abnormal, and 2 for normal. A mean score for each rat was calculated (maximum 2.0), and then the mean score of the cohort was calculated.

### 4.3. Sample Collection

Six weeks postsurgery, after the behavioural tests described above, all the animals were compassionately euthanised under carbon dioxide asphyxiation. Subsequently, samples (approximately 1 × 1.5 cm) of muscle (gastrocnemius muscle) and TLF were excised from the left (sciatic nerve injury) and right (contralateral) sides of the rats in the experimental group. The gastrocnemius muscle and TLF samples were also collected from the right side of the healthy rats in the control group. Each sample was divided into two pieces: one was preserved at −80 °C until the HA and collagen assays, and the other was fixed in 10% formalin for subsequent histological/immunohistochemical analysis. The procedures are demonstrated in Figure 9.

### 4.4. Quantification of Hyaluronan

A Purple-Jelley HA assay (Perchlorate-Free) (Biocolor Ltd., Carrickfergus, UK) was employed to determine the concentration of HA in skeletal muscle or fascia of rats. As previously described [22,54,55], 150 mg ± 50 mg of wet tissue (defrosted from −80 °C) was weighted and cut into small fragments using a surgical scalpel and transferred into 2.0 mL microcentrifuge tubes. Subsequently, samples were subjected to digestion in 400 μL of TRIS-HCl (50 mM, pH 7.6) with the addition of Proteinase K (Sigma) overnight at 55 °C to ensure complete enzymatic digestion. After centrifugation at 13,000× *g* for 10 min, the supernatants were carefully transferred into new microcentrifuge tubes and mixed with 1.0 mL of a GAG precipitation reagent. After 15 min, another round of centrifugation at 13,000× *g* for 10 min was performed to obtain the resulting residues. These residues were then dispersed in water and mixed with NaCl and cetylpyridinium chloride (CPC). After repeating the aforementioned steps and successfully recovering the total GAG content, HA isolation was performed by adding 500 μL of 98% ethanol to the samples, followed by centrifugation and full hydration in 100 μL of water. For the subsequent colorimetric analysis, 200 μL of a purple dye reagent was added to 20 μL aliquots of the test samples, standards, or reagent blanks. The absorbance value at 650 nm was measured using a Wallac Victor3 1420 Multilabel Counter (Perkin Elmer, Turku, Finland) in 96-microwell plates. To determine the concentration of HA in the samples, the absorbance values of the standard curve created with the HA standard (200 μg/mL) were converted into micrograms of HA contained in the total volume of 100 μL. After calculating the micrograms of HA extracted from the starting tissue samples, the average micrograms of HA per gram of wet tissue were determined for each sample (at least two measurements ± standard deviation for each sample).

### 4.5. Quantification of Collagen

A Total Collagen Assay Kit (Abcam Ltd., Cambridge, UK) was employed to determine the concentration of collagen in the gastrocnemius muscle and TLF of the rats. As previously described [56], 60–100 mg of wet tissue (defrosted from −80 °C) was weighted and cut into small fragments using a surgical scalpel. The fragments were then homogenised in distilled water at a ratio of 100 µL of water for every 10 mg of starting tissue. For each sample, 100 µL of homogenate was added to 100 μL of 10 N NaOH and heated at 120 °C for 1 h to facilitate collagen extraction. After cooling the vial on ice and neutralisation by adding 100 µL of 10 N HCl, the samples were centrifuged at 10,000 g for 5 min. From the centrifuged solution, 10 µL was transferred into a 96-well plate, which was then placed in an oven at 65 °C and evaporated for 10–20 min until white crystals formed. To each well, 100 µL of oxidising reagent (6 µL Chloramine T mixed + 94 µL oxidation buffer) was added and incubated at room temperature for 20 min. For the development of the colorimetric reaction, 50 µL of developer solution was added to each well at 37 °C for 5 min. Subsequently, 50 µL of concentrated dimethylaminobenzaldehyde (DMAB) was added, mixed thoroughly, and incubated on a plate heater at 65 °C for 45 min. Following incubation, absorbance was measured at 570 nm utilising a Wallac Victor3 1420 Multilabel Counter (Perkin Elmer, Finland), with data compared against a standard curve generated from collagen I standard solution (from 0 to 18 μg/mL), to obtain the micrograms of collagen per milligram of tissue. For each sample, at least two measurements were performed to obtain mean values ± standard deviation.

### 4.6. Histological Analysis 

Fixed samples (gastrocnemius muscle and TLF) were dehydrated in graded ethanol and in xylene and then embedded in paraffin. An amount of 5 µm slices were cut by microtome, then dewaxed and hydrated before proceeding with Picrosirius Red staining, Alcian Blue staining, and immunohistochemistry. 

Picrosirius Red staining: After deparaffination and rehydration, tissue sections were stained with Picrosirius Red solution for 13 min to selectively highlight collagen fibres. Subsequently, the sections were briefly rinsed in 0.01 M hydrochloric acid (HCl) for 20 s, then washed in deionised water quickly. After being dehydrated through an ascending alcohol series and cleared in xylene, the samples were finally mounted by Eukitt (Agar Scientific Elektron Technology, Stansted, UK)

Alcian Blue staining: Alcian Blue solution was used to stain acidic polysaccharides such as glycosaminoglycans (GAGs), according to the principle described by Scott and Dorling [57]. Dewaxed sections were incubated for 1 hr in sodium acetate buffer with pH 5.8 and MgCl_2_ 0.05 M and then stained for 2 h with 0.05% Alcian Blue in MgCl_2_ 0.05 M. Samples were then washed in 0.01 N HCl for 10 min and in distilled water [55]. The dehydrated samples were finally mounted by Eukitt (Agar Scientific).

Immunohistochemistry: Sections were treated with 0.5% H_2_O_2_ for 15 min to block endogenous peroxidases. After washing, the specimens were incubated for 1 h in 0.2% blocking solution (PBS + 0.2% Bovine Serum Albumin—BSA) and then incubated overnight at 4° C with anti-HABP (hyaluronic acid binding Protein, dilution 1:1000, Millipore Sigma-Aldrich, MI, Italy). After washing in PBS, the samples were incubated in HRP-conjugated Streptavidin (Jackson ImmunoResearch, Cambridgeshire, UK, dilution 1:250) for 30 min and then washed 3 times in PBS. The reaction was developed with 3,3′-diaminobenzidine (Liquid DAB + Substrate Chromogen System kit Dako, CA, USA) and stopped with distilled water. Slides were then dehydrated and mounted using Eukitt (Agar Scientific).

### 4.7. Image Analysis

All the images were acquired by Leica DMR microscope (Leica Microsystems, Wetzlar, Germany). 

For measuring the area percentage of collagen content (area %) of the gastrocnemius muscle, for each sample at least twenty images from two different tissue sections were taken at a 10× magnification after Picrosirius Red staining. Then the images were analysed by ImageJ software [58] (version 1.53) freely available at http://rsb.info.nih.gov/ij/ (accessed on 19 February 2024). The field area containing epimysium, perimysium, and endomysium was used to estimate the collagen level, normalised for the total area (the detailed procedure is described in Appendix A).

### 4.8. Statistical Analysis

Statistical analyses were performed by using IBM SPSS statistical software (version 25, SPSS, Chicago, IL, USA), and the significance level was set at 0.05. The Shapiro–Wilk test was used to assess normality, and Levene’s test was applied to evaluate the homogeneity of variances. All the data were reported as mean ± standard deviation (M ± SD). For normally distributed data, the differences in HA and collagen content in muscle and TLF among the left (sciatic nerve injury), right (contralateral), and control (healthy rats) groups were compared using ANOVA with the Tukey post hoc test (for normally distributed data with equal variances) or the Games–Howell post hoc test (for normally distributed data with unequal variances). For non-normally distributed data, the collagen area percentage was analysed using the Kruskal–Wallis test with post hoc Mann–Whitney U (MWU) tests.

## 5. Conclusions

This study reveals systemic ECM alterations following PNI, focusing on HA and collagen changes in lower limb muscles and the TLF in rats with sciatic nerve transection repaired by nerve grafts. It uniquely highlights significant HA and collagen changes in both the injured and contralateral limbs as well as the TLF. These findings underscore the need for a global perspective in analysing lower limb PNI, suggesting potential exacerbation of motor function impairment and increased risk of lower back dysfunction, emphasising comprehensive assessment and holistic rehabilitation. 

## Figures and Tables

**Figure 1 ijms-25-08945-f001:**
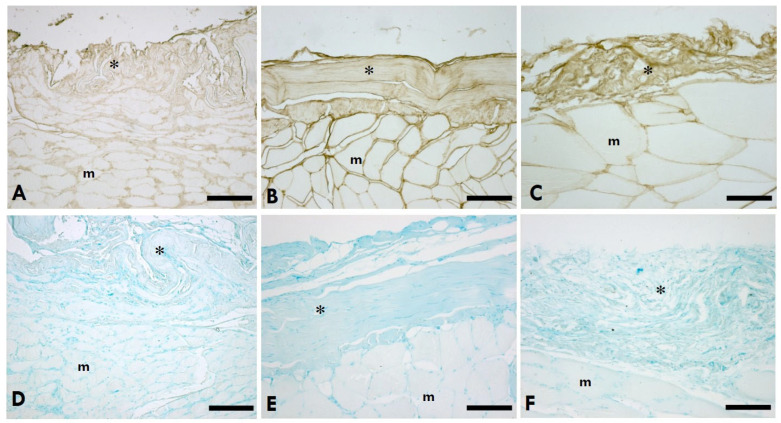
HA distribution in IMCT of the gastrocnemius muscle. (**A**–**C**): Anti-HABP (hyaluronic acid binding protein); (**D**–**F**): 0.05% Alcian Blue in MgCl_2_ 0.05 M; (**A**,**D**) = left (sciatic nerve injury) side; (**B**,**E**) = right (contralateral) side; (**C**,**F**) = control (healthy rats); * = IMCT; m = muscle. Scale bars: 150 µm.

**Figure 2 ijms-25-08945-f002:**
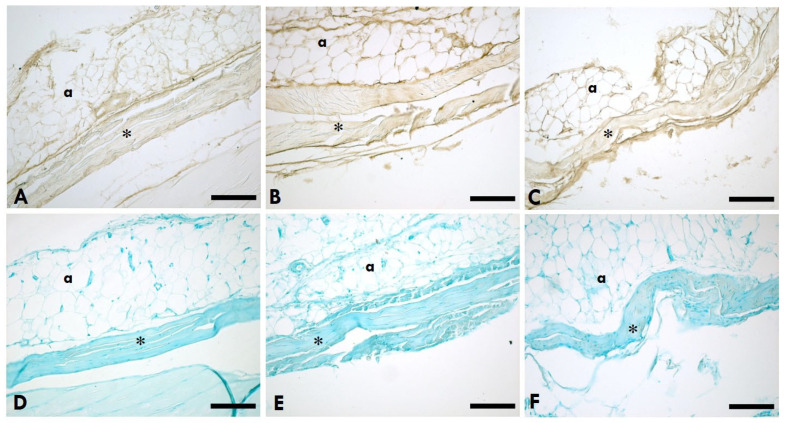
HA distribution in the TLF. (**A**–**C**): Anti-HABP (hyaluronic acid binding protein); (**D**–**F**): 0.05% Alcian Blue in MgCl_2_ 0.05 M; (**A**,**D**) = left (sciatic nerve injury) side; (**B**,**E**) = right (contralateral) side; (**C**,**F**) = control (healthy rats); * = TLF; a = adipocytes. Scale bars: 150 µm.

**Figure 3 ijms-25-08945-f003:**
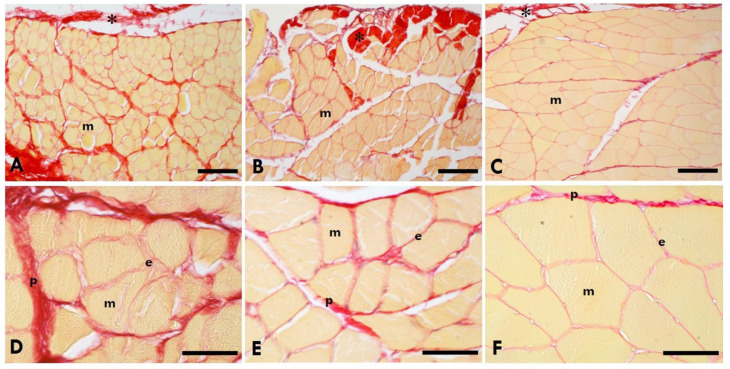
Picrosirius Red staining of the gastrocnemius muscle. (**A**,**D**) = Left (sciatic nerve injury) side; (**B**,**E**) = right (contralateral) side; (**C**,**F**) = control—healthy rats. * = IMCT; m = muscle; e = endomysium; p = perimysium. Scale bars: (**A**–**C**): 150 µm; (**D**–**F**): 50 µm. Collagen is in red, the muscle fibres of the gastrocnemius muscle in yellow.

**Figure 4 ijms-25-08945-f004:**
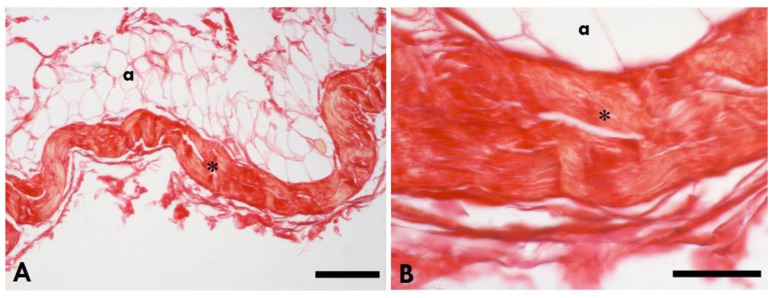
Picrosirius Red staining of the healthy TLF. (**A**): Scale bars 150 µm; (**B**): scale bars 50 µm; * = thoracolumbar fascia; a = adipocytes.

**Figure 5 ijms-25-08945-f005:**
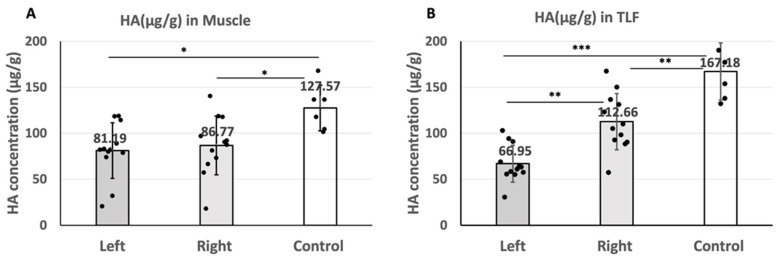
(**A**): The HA concentration in the gastrocnemius muscle on the left (sciatica nerve injury) side and right (contralateral) side of the left sciatic nerve injury rats and of the healthy rats (* *p* < 0.05). (**B**): The HA concentration in the TLF of the left (sciatica nerve injury) side and right (contralateral) side of the left sciatica nerve injury rats and of the healthy rats (** *p* < 0.01, *** *p* < 0.001). ANOVA with the Tukey post hoc test is applied to compare the difference between left and right sides of injured rats. Independent *t*-test is applied to compare the results of both sides of the injured rats and control separately.

**Figure 6 ijms-25-08945-f006:**
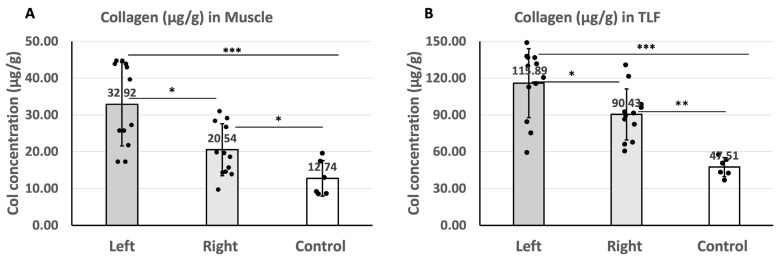
(**A**): The collagen concentration in the gastrocnemius muscle of the left (sciatic nerve injury) side and the right (contralateral) side of the left sciatic nerve injury rats and of the healthy rats (* *p* < 0.05, *** *p* < 0.01). (**B**): The HA concentration in the TLF of the left (sciatic nerve injury) side and the right (contralateral) side of the left sciatica nerve injury rats and of the healthy rats (* *p* < 0.05, ** *p* < 0.01, *** *p* < 0.001). ANOVA with the Tukey post hoc test (for collagen concentration in the muscle) and Games–Howell post hoc test (for collagen concentration in the TLF) are applied to compare the difference between left and right sides of injured rats. Independent *t*-test is applied to compare the results of both sides of the injured rats and control separately.

**Figure 7 ijms-25-08945-f007:**
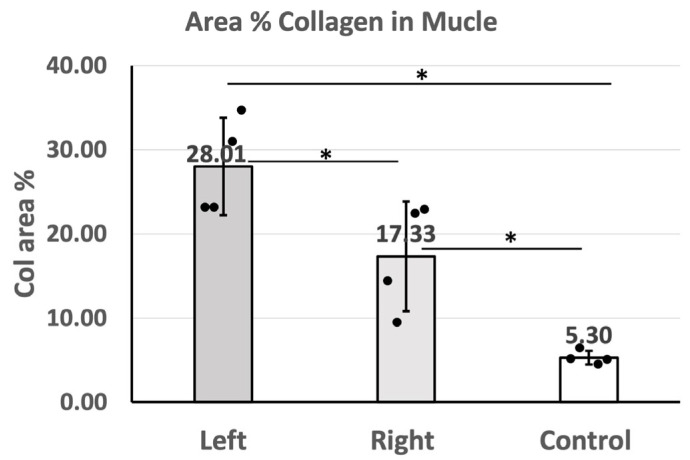
The area percentage of collagen content in the gastrocnemius muscle of the left (sciatica nerve injury) side (n = 4) and the right (contralateral) side (n = 4) of the left sciatic nerve injury rats and of the healthy rats in the control group (n = 4). Kruskal–Wallis test with post hoc Mann–Whitney test is applied to compare the difference between left and right sides of injured rats and control separately (* *p* < 0.05).

**Figure 8 ijms-25-08945-f008:**
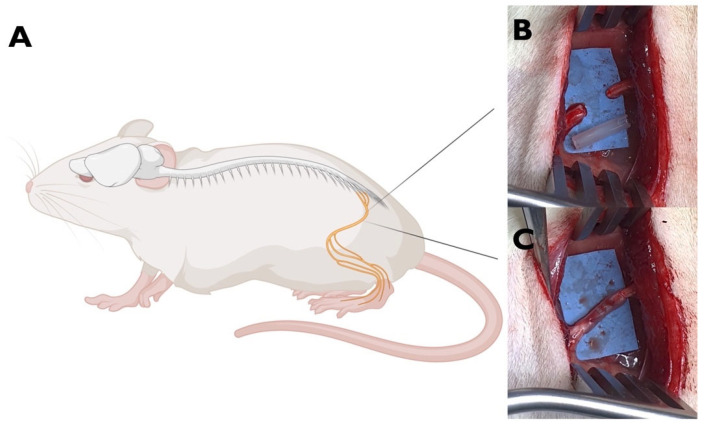
Animal model of PNI. (**A**) Left sciatic nerve of the rat (created with BioRender.com) (**B**) Nerve exposure and gap (5 mm) creation. (**C**). Nerve repaired by interposition of an OxPVA-based nerve graft (10 mm) between the proximal and the distal stumps.

**Figure 9 ijms-25-08945-f009:**
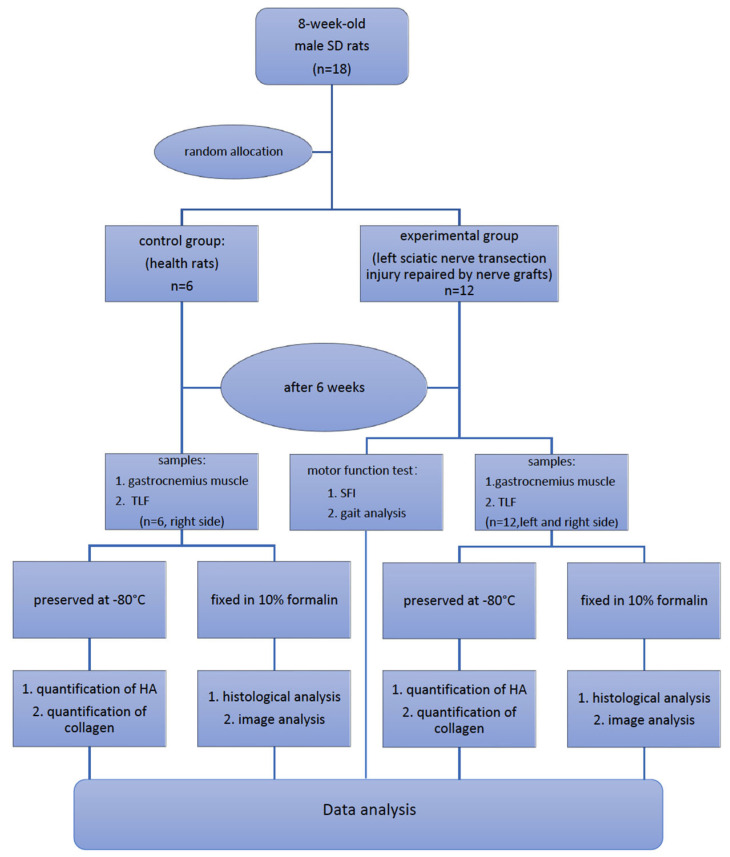
Experimental design.

**Table 1 ijms-25-08945-t001:** HA and collagen concentration and collagen area % in IMCT of the muscle and TLF.

Variable	L (n = 12)	R (n = 12)	Ctrl (n = 6)	*p*-Value L vs. R	*p*-Value L vs. Ctrl	*p*-Value R vs. Ctrl
µg HA/g muscle	81.19 ± 30.33	86.77 ± 31.96	127.57 ± 24.90	0.893	0.013 *	0.030 *
µg HA/g TLF	66.95 ± 20.08	112.66 ± 30.53	167.18 ± 31.13	0.001 **	0.000 ***	0.001 **
^#^ µg Col/g muscle	32.92 ± 11.34	20.54 ± 7.03	12.74 ± 4.83	0.01 *	0.000 ***	0.022 *
µg Col/g TLF	115.89 ± 28.18	90.43 ± 20.83	47.51 ± 7.82	0.027 *	0.000 ***	0.002 **
^$^ Col area% muscle	28.01 ± 5.78 (n = 4)	17.33 ± 6.52% (n = 4)	5.30 ± 0.82% (n = 4)	0.021 *	0.021 *	0.021 *

ANOVA with the Tukey post hoc test, ^#^ ANOVA with the Games–Howell post hoc test, ^$^ Kruskal–Wallis test with post hoc Mann–Whitney test. Values are presented as numbers or means ± SD. L: left (sciatic nerve injury) side, R: right (contralateral) side, Ctrl: control (healthy rats). * *p* > 0.05, ** *p* > 0.01, *** *p* > 0.001.

## Data Availability

Data contained within the article.

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
