# Peer review of "The Impact of Sciatic Nerve Injury on Extracellular Matrix of Lower Limb Muscle and Thoracolumbar Fascia: An Observational Study"

_ijms, 2024, doi:10.3390/ijms25168945_

Round 1

Reviewer 1 Report

Comments and Suggestions for Authors

The authors have presented important work that extends previous studies of the anatomical and functional impairments that occur following peripheral nerve injury. Using the rat model of sciatic nerve injury and conduit repair, the authors have studied quantitative changes in hyaluronan and collagen bilaterally, in gastrocnemius and thoracolumbar fascia, as well as alterations in the animals' gait, at the 6-week post-surgical time point. The authors have demonstrated both localized and systemic alterations in matrix components following peripheral nerve injury.

The Introduction contains a well-articulated summary of what is known of extracellular matrix components in muscle tissues and fascia following denervation injury. There is novelty in investigating the matrix components of the thoracolumbar fascia, as a fascial connection to the lower limb.

The goals of the study are well described. The Methods are described in enough detail to permit replication of the study. The biochemical analysis of levels of hyaluronan was critical, as the immunohistochemical staining for hyaluronan and histological staining for glycosaminoglycans appears similar across experimental, contralateral and control tissues (Figure 2).

The manuscript is well written and the Discussion links appropriately to extant literature. The study adds to our understanding of extracellular matrix changes in thoracolumbar fascia following peripheral nerve injury, and similar changes may be hypothesized to occur following spinal nerve injury in acute, and subsequently chronic, back pain.

Comments on the Quality of English Language

In addition to a few spelling corrections, three very minor edits are suggested:

Spell out sciatic functional index before first use of the abbreviation, SFI.

Move Figure 6 legend under Figure 6.

Revise the English grammar in the last sentence of the Discussion, page 8, lines 261-265.

Reviewer 2 Report

Comments and Suggestions for Authors

The manuscript titled "The Impact of Sciatic Nerve Injury on Extracellular Matrix of Lower Limb Muscle and Thoracolumbar Fascia: An Observational Study" presents a detailed study on the alterations in the concentration of hyaluronic acid (HA) and collagen in the gastrocnemius muscle and thoracolumbar fascia (TLF) in rats with unilateral sciatic nerve injury. The work offers significant insights into the systemic impact of peripheral nerve injuries on the extracellular matrix (ECM) and the functional consequences. However, there are several areas where the manuscript can be improved:

The abstract is comprehensive but could be more concise. Focus on the most important results and their implications. For instance, details about the experimental conditions can be abbreviated. Specify the main findings in terms of percentages or variations to provide a clearer idea.

The introduction could benefit from more recent references to highlight advancements in the field. The connection between peripheral nerve injuries and muscle alterations should be more explicitly stated to underline the study's significance. I recommend the authors clarify the rationale behind selecting the specific muscles (gastrocnemius and TLF) for this study.

Provide more details on the validation of the methods used, particularly for the techniques of HA and collagen quantification. Include a diagram or a flowchart of the experimental design for better visualization.

The discussion could better integrate the results with existing literature, emphasizing the novelty and implications of the findings. I suggest stating the study's limitations and proposing potential future research directions. The section discussing the potential mechanisms behind the increased expression of HA and ECM reorganization could be more detailed, considering alternative hypotheses and pathways.

The conclusion should be more concise and directly linked to the primary objectives and results of the study. Emphasize the broader implications of the findings for understanding muscle physiology and potential clinical applications.

Comments on the Quality of English Language

/

Reviewer 3 Report

Comments and Suggestions for Authors

The authors assess sciatic nerve injury induced hyaluronic acid reduction and collagen accumulation in gastrocnemius muscles and thoracolumbar fascia, which leads to exacerbation of motor function impairment upon injury.

The manuscript is well written and nicelly illustrated. There are several issues with data presentation and statistical analysis that I would suggest for authors to consider. The quantification diagrams on figures 5 and 6 are rather small and difficult to assess. Thus, I would suggest to make the graphs larger and, if possible, present them as dot plots, i.e. show single data points as well as average values and SD. Another point - authors use t tests for statistical analysis. Particularly, paired t test is used to compare left and right side. It is fine, although it is also possible to compare injured, uninjured and control animals using one way ANOVA, as a mor eellegant statistical test in this case. However, if the authors should stick to tests, they should properly report t statistics, with t values and degrees of freedom as well as exact p values.

Reviewer 4 Report

Comments and Suggestions for Authors

The manuscript presents a well-conducted animal model experiment on the impact of PNI on the local and distant muscles composition, and finds significant differences within injured, uninjured limb and in healthy controls.

With minor revisions to the methodology, discussion, and acknowledgment of limitations (the low number of subjects), the manuscript would be a great addition to the scientific literature.

Please provide additional details on the scoring criteria and reproducibility of the measurements to strengthen the reliability of the findings.

In the Discussion, provide a more in-depth exploration of the potential mechanisms underlying the observed ECM changes, and also the clinical implications of these findings for the treatment and rehabilitation of patients with PNI.

Comments on the Quality of English Language

English is fine in general.

Reviewer 5 Report

Comments and Suggestions for Authors

The Authors studied extracellular matrix (ECM) alterations affecting muscle function following the unilateral sciatic nerve injury in rats. The markers were hyaluronan (HA) and collagen concentration in gastrocnemius muscle and thoracolumbar fascia (TLF). Studies were performed on eighteen 8-week-old male Sprague-Dawley rats, which were divided into experimental (n=12 with left sciatic nerve transection injury, which was repaired)  and control (n=6) groups. The sample size was enough for the experiment's purposes. After six weeks, the motor function was evaluated. The histopathological and histochemical studies with modern available methods make the study very original, indicating motor structural and functional impairment and increased risk of low back dysfunctions. The study uniquely highlighted that nerve transection induced significant changes in HA and collagen concentrations within the muscle innervated by the injured nerve and in the muscles of the contralateral limb without nerve injury.

The high value of basic research in the presented work is undeniable.

The paper does not show the weak points; it is the example of the good and reliable scientific work.

Some flaws or corrections should be considered by the Authors as follows:

1. The first sentence in Abstract …”Peripheral nerve injury (PNI) is a complex clinical challenge affecting quality of life”… is obvious and its content should be closer to the study issue.

2. Delete numbers associated with keywords under Abstract.

3. Consider replacing "nerve conduits" (line 39 and in the Abstract) with "nerve grafts" or "reconstructive neurotubes" throughout the text.

4. line 41 – Consider replacing “of the end-organ” with “the effector’s end-plate”

5. line 64 – Consider replacing “in joint immobilization rats” with “rats with immobilized joints”

6. lines 88,89 – Consider deleting “potentially guiding future therapeutic strategies aimed at holistic functional recovery.” – brings nothing in this part of the text.

7. Provide more refs. for description of the footprint method. (4.2.2.); provide the table with comparison of the motor function results recorded on both sides and differences found between groups of animals (2.1). 

8. There is something wrong with lettering in Table 1.

9. Discussion – develop the possible clinical significance of the results.

10. The scheme of the refs. in the list is not in accordance with the MDPI style, refs. 1, 47 are mistaken.

Comments on the Quality of English Language

Minor corrections are required

Round 2

Reviewer 2 Report

Comments and Suggestions for Authors

The authors have thoroughly addressed all the reviewers' comments

Comments on the Quality of English Language

Moderate editing of English language required
